# In Vitro and In Vivo Anti-Inflammatory and Antidepressant-like Effects of *Cannabis sativa* L. Extracts

**DOI:** 10.3390/plants13121619

**Published:** 2024-06-12

**Authors:** Joonyoung Shin, Sangheon Choi, A Yeong Park, Suk Ju, Bitna Kweon, Dong-Uk Kim, Gi-Sang Bae, Dongwoon Han, Eunjeong Kwon, Jongki Hong, Sungchul Kim

**Affiliations:** 1Institute for Global Rare Disease Network, Professional Graduate School of Korean Medicine, Wonkwang University, Iksan 54538, Republic of Korea; spm1219@naver.com (J.S.); heoniappa1@gmail.com (S.C.); iam-mamas@naver.com (A.Y.P.); christineju@hanmail.net (S.J.); dwhan@hanyang.ac.kr (D.H.); 2Department of Pharmacology, School of Korean Medicine, Wonkwang University, Iksan 54538, Republic of Korea; kbn306@naver.com (B.K.); ckck202@naver.com (D.-U.K.); baegs888@wku.ac.kr (G.-S.B.); 3Research Center of Traditional Korean Medicine, Wonkwang University, Iksan 54538, Republic of Korea; 4Department of Global Health and Development, Hanyang University, Seoul 04763, Republic of Korea; 5College of Pharmacy, Kyung Hee University, Seoul 02447, Republic of Korea; dmswjd4876@naver.com (E.K.); jhong@khu.ac.kr (J.H.)

**Keywords:** *Cannabis sativa* L., lipopolysaccharide, entourage effect, anti-inflammatory effect

## Abstract

*Cannabis sativa* L. has been widely used by humans for centuries for various purposes, such as industrial, ceremonial, medicinal, and food. The bioactive components of *Cannabis sativa* L. can be classified into two main groups: cannabinoids and terpenes. These bioactive components of *Cannabis sativa* L. leaf and inflorescence extracts were analyzed. Mice were systemically administered 30 mg/kg of *Cannabis sativa* L. leaf extract 1 h before lipopolysaccharide (LPS) administration, and behavioral tests were performed. We conducted an investigation into the oxygen saturation, oxygen tension, and degranulation of mast cells (MCs) in the deep cervical lymph nodes (DCLNs). To evaluate the anti-inflammatory effect of *Cannabis sativa* L. extracts in BV2 microglial cells, we assessed nitrite production and the expression levels of inducible nitric oxide synthase (iNOS), cyclooxygenase (COX)-2, interleukin (IL)-1β, IL-6, and tumor necrosis factor (TNF)-α. The main bioactive components of the *Cannabis sativa* L. extracts were THCA (a cannabinoid) and β-caryophyllene (a terpene). *Cannabis sativa* L. leaf extract reduced the immobility time in the forced swimming test and increased sucrose preference in the LPS model, without affecting the total distance and time in the center in the open field test. Additionally, *Cannabis sativa* L. leaf extract improved oxygen levels and inhibited the degranulation of MCs in DCLNs. The *Cannabis sativa* L. extracts inhibited IL-1β, IL-6, TNF-α, nitrite, iNOS, and COX-2 expression in BV2 microglia cells. The efficacy of *Cannabis sativa* L. extracts was suggested to be due to the entourage effect of various bioactive phytochemicals. Our findings indicate that these extracts have the potential to be used as effective treatments for a variety of diseases associated with acute inflammatory responses.

## 1. Introduction

*Cannabis sativa* L. is an annual dioecious plant that has been cultivated mainly in Central Asia (India and China) since ancient times [1]. For centuries, *Cannabis sativa* L. has been widely utilized by humans for various purposes, including rituals [1], industry [2], pharmaceutical applications [3], and food [4]. One emerging use of Cannabis in food products is in the form of Cannabis protein, Cannabis seeds, Cannabis seed oil, and Cannabis-based milk [5]. In the United States, a variety of snack foods containing cannabinoids such as cannabidiol (CBD) are available for purchase, including gummies [6], sodas [7], and snack bars [8]. Additionally, Cannabis flour has been utilized to optimize the formulations of various food products such as Indian flatbread (Chapati), pasta, bread, and cookies [9,10,11]. The production of new food products using Cannabis is experiencing a promising growth trend. However, because Cannabis contains various bioactive phytochemicals, it is generally necessary to comply with regulations when incorporating Cannabis into foods [12]. The bioactive phytochemicals found in cannabis can be categorized into two main groups: cannabinoids and terpenes. Cannabinoids are interesting compounds found in *Cannabis sativa* L., including the psychoactive Δ9-tetrahydrocannabinol (Δ9-THC) and the non-psychoactive CBD [4]. Δ9-THC and CBD bind to endocannabinoid receptors [13] and have therapeutic effects on epilepsy, pain, anti-inflammatory conditions, major depressive disorder, and drug addiction [14,15,16]. And terpenes are one of the largest and most diverse groups in plants [17]. The major terpenes are α-pinene, β-pinene, myrcene, limonene, β-caryophyllene, and α-humulene. Although terpenes are primarily responsible for their characteristic aroma, they also have beneficial health benefits such as anticancer, antibacterial, antifungal, antiviral, analgesic, anti-inflammatory, and antiparasitic activities [18]. In particular, treatment with β-caryophyllene has been found to have a range of beneficial effects on neurological diseases, including anticonvulsant, analgesic, muscle relaxant, sedative, and antidepressant properties [19].

Lipopolysaccharide (LPS) is a crucial bacterial component that contributes significantly to the development of various chronic diseases. [20]. The administration of LPS not only induces sickness behavior but also leads to a depressive-like state, anhedonia, and anxiety-like behavior in animals [21,22]. Moreover, the LPS stimulation of BV2 microglial cells triggers the expression of inflammatory mediators [inducible nitric oxide synthase (iNOS), cyclooxygenase (COX)-2] and pro-inflammatory cytokines [interleukin (IL)-1β, IL-6, and tumor necrosis factor (TNF)-α] [23]. Surprisingly, mice administered intraperitoneally with LPS (5 mg/kg) exhibited impaired deep cervical lymph nodes (DCLNs) and meningeal lymphatic drainage and morphology, resulting in exacerbated neuroinflammation and cognitive impairment [24]. To date, the histological characteristics of mast cells (MCs) in DCLNs, which are one of the important regulators of inflammation and immune responses after LPS administration, have not been studied.

In this study, we investigated the effects of *Cannabis sativa* L. leaf extract on depressive-like behavior in an animal model of LPS-induced depression. We assessed changes in oxygen indices and MCs in DCLNs in this model. Additionally, we evaluated the effects of *Cannabis sativa* L. leaf and inflorescence extracts on various inflammatory markers, such as pro-inflammatory cytokines and inflammatory mediators, in BV2 microglial cells.

## 2. Results

### 2.1. Cannabinoid and Terpene Contents in Cannabis sativa L. Leaf and Inflorescence Extracts

A typical total ion chromatogram (TIC) for the conversion products of *Cannabis sativa* L. leaf and inflorescence extracts under acidic reaction conditions, as analyzed in gas chromatography–mass spectrometry (GC-MS) scan mode, is shown in Figure 1A,B. THCA (peak 6, 10.89 mg/g) and CBCA (peak 7, 1.96 mg/g) were detected as major cannabinoid components of *Cannabis sativa* L. leaf extract (Figure 1A). Also, THCA (peak 6, 117.67 mg/g), CBCA (peak 7, 2.44 mg/g), and CBGA (peak 8, 12.17 mg/g) were detected as major cannabinoid components of *Cannabis sativa* L. inflorescence extract (Figure 1B).

A total of 26 terpenes were detected in the leaf and inflorescence samples using headspace gas chromatography/mass spectrometry (HS-GC/MS) (Figure 1C,D). β-caryophyllene (peak 8, 3455.65 µg/g), α-humulene (peak 12, 178.73 µg/g), and limonene (peak 4, 74.22 µg/g) were detected as major terpenes of *Cannabis sativa* L. leaf extract (Figure 1C). Additionally, myrcene (peak 3, 11,159.79 µg/g), limonene (peak 4, 1157.78 µg/g), and β-caryophyllene (peak 8, 3397.04 µg/g) were detected as major terpenes of *Cannabis sativa* L. inflorescence extract (Figure 1D). In particular, β-caryophyllene was detected at a high level in both the *Cannabis sativa* L. leaf and inflorescence extracts.

### 2.2. Antidepressant-like Effect of Cannabis sativa L. Leaf Extract in the LPS Mouse Model

In the sucrose preference test (SPT), the LPS group presented a decreased sucrose preference (53.0 ± 5.9%) compared to the control group (81.6 ± 4.8%, *p* < 0.001). The administration of the *Cannabis sativa* L. leaf extract to LPS mice (65.2 ± 4.4%) increased their sucrose preference compared to the LPS group (53.0 ± 5.9%, *p* < 0.01) (Figure 2B). In the forced swimming test (FST), the LPS group presented an increased immobility time (118.2 ± 20.2 s) compared to the control group (69.2 ± 15.2 s, *p* < 0.01). The administration of the *Cannabis sativa* L. leaf extract prevented the increase in immobility time induced by LPS (86.8 ± 13.7 s, *p* < 0.05, Figure 2C). The LPS group traveled a shorter distance in the open field test (OFT) (9.15 ± 2.8 s) compared to the control group (19.4 ± 1.7 s, *p* < 0.001). However, the administration of the *Cannabis sativa* L. leaf extract to the LPS mice did not significantly change the distance traveled (11.4 ± 2.7 s, Figure 2D). In addition, the LPS group spent less time in the center (9.1 ± 10.1 s) compared to the control group (46.9 ± 11.2 s, *p* < 0.001). However, the *Cannabis sativa* L. leaf extract administration to the LPS mice did not significantly affect their time in the center when compared to the LPS group (14.7 ± 6.8 s, Figure 2E). Representative tracks from the mice in the OFT are shown in Figure 2F. The control group was more active compared to the other groups. Compared to other behavioral experiments, the administration of *Cannabis sativa* L. leaf extract had no significant effect on the behavioral parameters examined in the OFT.

### 2.3. Improved Oxygen Indicators of Cannabis sativa L. Leaf Extract in LPS Mouse Model

The LPS group exhibited a decrease in arterial oxygen saturation (87.4 ± 2.6%) compared to the control group (95.6 ± 1.7%, *p* < 0.001). The administration of the *Cannabis sativa* L. leaf extract prevented the decrease in arterial oxygen saturation caused by LPS (91.6 ± 1.3 s, *p* < 0.05). The oxygen saturation of the LPS group had a relatively high standard deviation and was unstable during the 5 min measurement period (Figure 3A).

To measure the oxygen tension in DCLNs, an incision was made in the skin and subcutaneous fat tissues around the neck. To stabilize the position of DCLNs, the sternocleidomastoid muscle was secured using forceps. The LPS group demonstrated a decrease in oxygen tension in DCLNs (11.8 ± 1.9 mmHg) compared to the control group (26.4 ± 3.3 mmHg, *p* < 0.001). The administration of the *Cannabis sativa* L. leaf extract prevented the decrease in oxygen tension induced by LPS (21.7 ± 3.1 mmHg, *p* < 0.001, Figure 3B).

### 2.4. Effects of Cannabis sativa L. Leaf Extract on MCs in DCLNs

In MCs of DCLNs, there was no significant difference in the DCLN area in all the groups [1.17 ± 0.29 ${mm}^{2}$ (control), 1.12 ± 0.12 ${mm}^{2}$ (LPS), 1.18 ± 0.36 ${mm}^{2}$ (*Cannabis sativa* L. leaf extract + LPS), respectively; Figure 4A]. The LPS group showed an increased number of MCs (6.4 ± 2.79) compared to the control group (2.4 ± 1.51, *p* < 0.05). However, the administration of the *Cannabis sativa* L. leaf extract did not significantly change the MC number (5.8 ± 1.30, Figure 4B). The LPS group showed an increased MC degranulation rate (59.2 ± 14.18%) compared to the control group (5.0 ± 11.18%, *p* < 0.001). *Cannabis sativa* L. leaf extract administration inhibited the degranulation ratio of MCs induced by LPS (30.4 ± 11.49%, *p* < 0.01, Figure 4C).

### 2.5. Effect Cannabis sativa L. Leaf and Inflorescence Extracts on Cell Viability and Nitrite Oxide (NO) Production in LPS-Stimulated BV2 Microglial Cells

In order to determine the concentration range of *Cannabis sativa* L. leaf and inflorescence extracts in BV2 microglial cells, we performed the MTT assay to evaluate the cytotoxicity of the *Cannabis sativa* L. leaf and inflorescence extracts. BV2 microglial cells were treated with the *Cannabis sativa* L. leaf and inflorescence extracts at concentrations of 200, 400, 800, 1600, and 3200 µg/mL for 24 h. Up to a dose of 1600 µg/mL, the *Cannabis sativa* L. leaf and inflorescence extracts did not affect cell viability (Figure 5A,C). Therefore, we chose a concentration range of *Cannabis sativa* L. leaf and inflorescence extracts of 0 to 1600 µg/mL for further experiments.

In order to investigate the anti-inflammatory effect of the *Cannabis sativa* L. leaf and inflorescence extracts, we measured nitrite production using the Griess assay. BV2 microglial cells were pretreated with the *Cannabis sativa* L. leaf and inflorescence extracts at concentrations of 200, 400, 800, and 1600 µg/mL for 1 h and then stimulated with LPS for 24 h. Our data showed that the LPS treatment markedly induced nitrite production in BV2 microglial cells. However, the increase in nitrite by LPS was significantly decreased by the *Cannabis sativa* L. leaf and inflorescence extracts in a concentration-dependent manner (Figure 5B,D).

### 2.6. Inhibitory Effect of Cannabis sativa L. Leaf and Inflorescence Extracts on iNOS and COX-2 Expression in LPS-Stimulated BV2 Microglial Cells

To investigate inflammatory mediators such as iNOS and COX-2, we performed Quantitative Real-Time Reverse Transcription Polymerase Chain Reaction (RT-PCR). BV2 microglial cells were pretreated with the *Cannabis sativa* L. leaf and inflorescence extracts at concentrations of 200, 400, 800, and 1600 µg/mL for 1 h and then stimulated with LPS for 6 h. Our data showed that LPS treatment markedly increased the mRNA levels of iNOS and COX-2 in BV2 microglial cells. The *Cannabis sativa* L. leaf extract effectively suppressed the mRNA levels of COX-2 and iNOS, particularly at 1600 µg/mL (Figure 6A,B). The *Cannabis sativa* L. inflorescence extract demonstrated a concentration-dependent suppression of COX-2 and iNOS mRNA levels (Figure 6C,D).

### 2.7. Inhibitory Effect of Cannabis sativa L. Leaf and Inflorescence Extracts on IL-1β, IL-6, and TNF-α Expression in LPS-Stimulated BV2 Microglial Cells

To investigate pro-inflammatory cytokines such IL-1β, IL-6, and TNF-α, we performed the Quantitative RT-PCR. BV2 microglial cells were pretreated with the *Cannabis sativa* L. leaf and inflorescence extracts at concentrations of 200, 400, 800, and 1600 µg/mL for 1 h and then stimulated with LPS for 6 h. Our data showed that LPS treatment markedly increased the mRNA levels of IL-1β, IL-6, and TNF-α in BV2 microglial cells. The mRNA levels of IL-1β, IL-6, and TNF-α were suppressed in a concentration-dependent manner by the *Cannabis sativa* L. leaf extract. This suppression was particularly effective at a concentration of 1600 µg/mL (Figure 7A–C). Similarly, the *Cannabis sativa* L. inflorescence extract also suppressed the mRNA levels of IL-1β and TNF-α, especially at a concentration of 1600 µg/mL (Figure 7D–F).

## 3. Discussion

In this study, we showed that acute *Cannabis sativa* L. leaf extract induced an antidepressant-like effect in a model of depression induced by LPS administration. Furthermore, it was shown that *Cannabis sativa* L. leaf extract restored arterial oxygen saturation and oxygen tension while reducing the degranulation ratio of MCs in DCLNs. Additionally, the anti-inflammatory effect of the *Cannabis sativa* L. leaf and inflorescence extracts was demonstrated through a decrease in the expression of inflammatory mediators (iNOS, COX-2) and pro-inflammatory cytokines (IL-1β, IL-6, and TNF-α) in LPS-stimulated BV2 microglial cells.

The administration of LPS systemically led to a state similar to depression, which can be observed through the increased occurrence of behavioral despair and anhedonia [20,21,25,26]. In our study, there were no statistically significant correlations between *Cannabis sativa* L. leaf extract administration and the changes in observed movements, such as total distance and time in the center in the OFT. Acute CBD administration also did not show significant changes in the OFT, suggesting that acute *Cannabis sativa* L. leaf extract administration did not change depression-like behavior in the OFT [16,27]. These results led us to plan further behavioral experiments following the chronic administration of *Cannabis sativa* L. leaf extract. The reduction in immobility time in the FST due to *Cannabis sativa* L. leaf extract administration contrasts with the lack of effect on movement observed in the OFT, supporting the effect of *Cannabis sativa* L. leaf extract on depressive-like behavior. Additionally, the anhedonic state induced by LPS was partially prevented by the *Cannabis sativa* L. leaf extract, supporting the alleviation of various depressive symptoms observed in this model, consistent with the results obtained using CBD and Δ9-THC [16,27,28,29]. Additionally, the administration of β-caryophyllene, one of the main components of terpenes, shows antidepressant and anti-inflammatory properties through behavioral experiments such as the FST, the OFT, etc., and the inhibition of cytokine expression [30,31,32]. The anti-inflammatory effects of β-caryophyllene have been shown to be primarily provided through the cannabinoid receptor 2 [32]. Until now, modern clinical research has primarily concentrated on exploring the pharmacological potential of individual purified cannabinoids, like CBD and Δ9-THC, when studying the medicinal properties of cannabis substances. Nevertheless, numerous publications over the years have demonstrated that treatments based on the entire cannabis plant tend to yield superior therapeutic effects compared to drugs composed solely of purified cannabinoids [33,34,35,36,37]. In our study, it was confirmed that *Cannabis sativa* L. leaf extract contains various bioactive components, such as THCA, CBCA, β-caryophyllene, and α-humulene. Therefore, we propose that the antidepressant-like effects of *Cannabis sativa* L. leaf extract are attributed to “the entourage effect”, a synergistic interaction between different bioactive components in cannabinoids and terpenes.

LPS administration not only causes depression but also activates the innate immune response, leading to the secretion of various inflammatory cytokines such as IL-6, IL-1β, and TNF-α, causing neuroendocrine and neurochemical changes [25]. Also, LPS administration increases the arterial partial pressure of carbon dioxide, reduces the arterial partial pressure of oxygen, arterial oxygen saturation, the HCO3(-) concentration, and pH, and increases LWCI and MPO activities [38,39]. In our study, we observed that arterial oxygen saturation, which had been reduced by LPS-induced inflammation, was significantly increased upon *Cannabis sativa* L. leaf extract administration. Furthermore, the administration of the *Cannabis sativa* L. leaf extract resulted in a notable increase in DCLNs’ oxygen tension. DCLNs are directly connected to meningeal lymphatic vessels and serve as drains for macromolecules, cellular waste, toxic substances, and immune cells [40]. Since the discovery of meningeal lymphatics in 2015, there has been a growing body of evidence that shows the crucial role of the meningeal lymphatic system in regulating immune responses and inflammation in the central nervous system [40,41,42,43,44,45,46]. Moreover, recent studies suggest that the drainage of meningeal lymphatics driven by VEGF-C plays a crucial role in regulating depression-like behavior and facilitating a robust immune response against brain tumors [47,48,49]. In our study, LPS administration increased the number of MCs in DCLNs. This can be inferred because LPS directly activates MCs through Toll-like receptors (TLRs) to secrete various factors, including cytokines and growth factors [50]. In particular, TLR4 can be activated by LPS, subsequently stimulating the production and release of multiple cytokines by MCs and surrounding tissues, as well as the release of preformed granules, through an MC/histamine/NF-κB-dependent pathway [51]. Furthermore, although MCs are commonly found in lymph nodes, their population significantly expands during inflammation. In this state, they play an active role in attracting immune cells to the lymph nodes by releasing cytokines and chemokines [52,53,54]. In our study, the number of MCs in DCLNs, which was increased by LPS administration, was not decreased by *Cannabis sativa* L. leaf extract administration. Nevertheless, it was found that the degranulation ratio decreased significantly, indicating its potential for reducing the LPS-induced inflammatory response. This led us to plan future experiments to demonstrate that the regulation of MC activity within DCLNs is involved in draining connected meningeal lymphatic vessels and regulating immune and inflammatory responses.

When the inflammatory response of BV2 microglial cells is excessive and uncontrolled, it results in an increased secretion of inflammatory mediators such as iNOS and COX-2, as well as pro-inflammatory cytokines like IL-1β, IL-6, and TNF-α. This, in turn, contributes to the pathological conditions of neurological diseases [55]. Therefore, a promising approach for treating neurological diseases is to regulate the levels of inflammatory mediators and pro-inflammatory cytokines. NO is produced through the oxidation of nitrite and has a crucial role in physiological synthesis. Its regulation during inflammation is primarily controlled by iNOS [56]. However, the excessive overexpression of iNOS results in the overproduction of NO, ultimately leading to cellular injury and inflammation [57,58]. Similar to iNOS, COX-2 is a substance that can be induced by external stimuli like LPS, cytokines, and chemokines. Its expression increases in response to these stimuli, and the excessive production of COX-2 can lead to inflammation [59]. In this study, we assessed the cytotoxic and anti-inflammatory effects of *Cannabis sativa* L. leaf and inflorescence extracts in BV2 microglial cells induced with LPS, which is a model of cellular brain inflammation. Additionally, we verified the expression levels of NO, iNOS, and COX-2 as inflammatory markers. The findings revealed that the treatment with the *Cannabis sativa* L. leaf and inflorescence extracts significantly inhibited LPS-induced nitrite production and the increased expression of iNOS and COX-2 in microglia cells. This indicates that *Cannabis sativa* L. leaf and inflorescence extracts have the ability to reduce the production of inflammatory mediators.

When inflammatory cells are activated, excessive secretion of not only NO but also inflammatory cytokines, such as IL-1β, IL-6, and TNF-α, can worsen the inflammatory response [60]. In this study, we evaluated the inhibitory effect of *Cannabis sativa* L. leaf and inflorescence extracts on the expression of the pro-inflammatory cytokines IL-1β, IL-6, and TNF-α as additional markers of inflammation. The results showed that the *Cannabis sativa* L. leaf extract effectively inhibited the increase in IL-1β, IL-6, and TNF-α expression induced by LPS in BV2 microglial cells in a concentration-dependent manner. Additionally, in the *Cannabis sativa* L. inflorescence extract, inhibition was observed at concentrations of 800 and 1600 μg/mL. Therefore, the concentration that effectively suppressed the mRNA levels of pro-inflammatory cytokines (IL-1β, IL-6, and TNF-α) and inflammatory mediators (iNOS and COX-2) in both extracts was 1600 μg/mL. These results suggest that *Cannabis sativa* L. leaf and inflorescence extracts may effectively reduce the production of pro-inflammatory cytokines and inflammatory mediators in response to LPS.

There are several limitations to our study. First, the precise mechanisms underlying the antidepressant-like and anti-inflammatory effects of *Cannabis sativa* L. leaf and inflorescence extracts were not fully elucidated. Future experiments, including Western blotting and immunofluorescence, are necessary to investigate these mechanisms in detail. Second, the study primarily relied on the forced swim test and the sucrose preference test to assess behavioral changes, which may not comprehensively capture all aspects of depressive and anxiety-like behaviors. Additional behavioral tests, such as the tail suspension test and elevated plus maze, could provide further insights. Third, the study did not include a detailed toxicity assessment of the *Cannabis sativa* L. extracts, which is crucial for evaluating the safety profile of potential therapeutic agents. Finally, the sample size for the in vivo experiments was relatively small (*n* = 5 per group), which may limit the generalizability of the findings. Larger-scale studies are required to confirm these results and further explore the therapeutic potential of *Cannabis sativa* L. extracts.

## 4. Materials and Methods

### 4.1. Animals

Male C57BL/6 mice aged 8 weeks (22–24 g) were obtained from Samtaco BIO (Gyeonggi-do, Republic of Korea). Mice were group-housed with 5 animals per cage and had free access to food and water at room temperature, with a constant 12:12 h light–dark cycle. All animal experiments were carried out in accordance with the guidelines approved by the Wonkwang University Animal Experiment Ethics Committee (WKU22-125). Anesthesia was induced by intramuscular injection into the hindlimb with an anesthetic cocktail containing Zoletil (20 mg/kg, Virbac, Carros, France) and xylazine (5 mg/kg, Bayer, Leverkusen, Germany). All mice were randomly divided into the following three groups, each containing five mice: (i) the control group (vehicle treatment), (ii) the LPS group (treated with LPS only), and (iii) the *Cannabis sativa* L. group (administered with 30 mg/kg of *Cannabis sativa* L. leaf extract 1 h before the administration of 500 µg/kg of LPS).

### 4.2. Drugs

*Cannabis sativa* L. leaf and inflorescence (white widow hybrid) were received from Nongboomind Company (Seoul, Republic of Korea) under the permission of the Ministry of Food and Drug Safety. *Cannabis sativa* L. leaf and inflorescence were refluxed with ethanol (96%) in a 1:5 (*w*/*v*) sample-to-solvent ratio at 85–90 °C for 2 h. The process was repeated three times, and the obtained liquid extract was filtered using filter paper (no. 20; Hyundai Micro, Seoul, Republic of Korea). The obtained filtrate was evaporated using a rotary evaporator to concentrate the extract, yielding a viscous residue, which was then freeze-dried. The completely dried extract was dissolved in distilled water and filtered to obtain the fractions.

The identification and quantification of cannabinoids in *Cannabis sativa* L. leaf and inflorescence extracts were carried out using GC-MS, as previously reported [61]. The analyzed cannabinoids were a reference standard mixture of eight neutral cannabinoids (purity ≥ 99.0%) [cannabidivarin (CBDV), tetrahydrocannabivarin (THCV), cannabidiol (CBD), cannabinol (CBN), delta-9-tetrahydrocannabinol (Δ9-THC), delta-8-tetrahydrocannabinol (Δ8-THC), cannabichromene (CBC), cannabigerol (CBG)] and a standard mixture of six acidic cannabinoids (purity ≥ 98.5%) [cannabichromenic acid (CBCA), cannabidivarinic acid (CBDVA), cannabidiolic acid (CBDA), cannabigerolic acid (CBGA), tetrahydrocannabivarinic acid (THCVA), and tetrahydrocannabinolic acid-A (THCA-A)], and isotopically labeled internal standards such as Δ9-THC-d3, CBD-d3, CBDA-d3, and THCA-d3 (purity ≥ 99.9%). And HS-GC/MS was employed to identify the major terpenes in the *Cannabis sativa* L. leaf and inflorescence extracts [62]. Quantitative analysis was performed on α- and β-pinene, myrcene, limonene, β-caryophyllene, α-humulene, etc., which are known as the major terpenes in cannabis, in order to investigate terpenes.

LPS (Sigma-Aldrich, Darmstadt, Germany) was dissolved in saline and used at a dose of 500 μg/kg [25,26]. Both *Cannabis sativa* L. leaf extract and LPS were administered intraperitoneally. Animals that did not receive drugs were injected with the corresponding vehicle (control group). In vitro, LPS (1 µg/mL) was treated for 24 h in nitrite oxide concentration experiments and for 6 h in RT-PCR experiments [63].

### 4.3. In Vivo Experimental Design

A graphical representation of the experimental procedure is shown in Figure 2A. Eight days prior to conducting the experiment, the oxygen saturation levels of the animals in all groups were measured. The following day, the animals were individually placed in their own habitat and allowed to acclimate to a choice between a 2% sucrose solution and water. *Cannabis sativa* L. leaf extract (30 mg/kg) was administered 1 h before LPS administration (500 μg/kg). The OFT and the FST were performed 24 h after LPS administration. Sucrose preference was assessed during the 24 h period. Two hours after completing the behavioral test, arterial oxygen saturation and oxygen tension in DCLNs were measured following the administration of anesthesia. Subsequently, DCLNs were collected for the histological observation of MCs. Samples were promptly collected and stored in 10% neutral buffered formalin (Sigma-Aldrich, Seoul, Republic of Korea) until needed. Animals used for MC staining of DCLNs were sacrificed by cervical dislocation.

### 4.4. Sucrose Preference Test (SPT)

This test aimed to evaluate anhedonia. The animals were housed individually and given access to water and a 2% sucrose solution for one week prior to the experiment. This allowed them to become accustomed to having both options available, as previously reported [64]. To prevent any potential position bias, the two bottles were switched daily. The sucrose preference was determined by calculating the percentage of sucrose solution consumed compared to the total amount of liquid intake during the 24 h period following LPS administration.

### 4.5. Open Field Test (OFT)

We used an open field apparatus to assess depressive-like behaviors in mice, following previously established protocols [16]. The OFT utilized an empty square arena (40 × 40 × 40 cm) made of plastic with a white base. The central region measured 20 × 20 cm. Mice were individually placed in one corner of the OFT apparatus, and their spontaneous activity was recorded for 6 min using a camcorder (FDR-AX43A, SONY, Tokyo, Japan). Data analysis was conducted at the 5 min point, excluding the initial minute, using the Tracker video analysis and modeling tool (USA). The parameters assessed included total distance (m) and time in the center (s). After each test, the open field apparatus was cleaned with 70% ethanol to eliminate any odor cues.

### 4.6. Forced Swimming Test (FST)

The FST was conducted following established procedures [65]. The test apparatus comprised a transparent 10 L beaker (diameter 210 mm, height 350 mm) filled halfway with water (maintained at 23 °C to 25 °C) and equipped with a webcam (APC930U, ABKO, Seoul, Republic of Korea) positioned in the front. The immobility time of the mice (s) was recorded for a total of 6 min, and data analysis was performed at the 4 min point, excluding the initial 2 min.

### 4.7. Measurement of Arterial Oxygen Saturation

The arterial oxygen saturation in mice under anesthesia was measured non-invasively using the MouseOx^®^ Pulse Oximeter (Starr Life Sciences Corp., Oakmont, PA, USA) [66]. The sensor used was a throat sensor with relatively excellent signal stability. Oxygen saturation was measured in all groups before any drug treatment. After the behavioral tests, each animal was measured every minute for a total of 5 min. Generally, the signal is irregular for about 3 min after the sensor is attached to the neck, so oxygen saturation values were collected once the oxygen saturation, heart rate, respiratory expansion, and pulse expansion indicators were all stabilized.

### 4.8. Measurement of Oxygen Tension

Oxygen tension was measured with an oxygen-only bare-fiber sensor (tip diameter 250 µm, OxyLite Pro, Southampton, UK) [67]. To obtain local pO_2_ values, an incision was made in the neck skin and subcutaneous fat under anesthesia. The DCLNs beneath the sternocleidomastoid muscle were then measured for their pO_2_ values. The pO_2_ needle sensor was meticulously positioned above each node location, and the sensor’s current signal, which was directly proportional to the pO_2_ value, was continuously monitored. Each procedure was repeated three times to ensure the accuracy of the sensor measurements.

### 4.9. Staining of Mast Cells (MCs)

The DCLNs were first fixed in 10% neutral buffered formalin overnight and then underwent standardized histological procedures to create tissue slides. Subsequently, toluidine blue staining was performed to verify the area, number of MCs, and degranulation of the DCLNs [68]. The area and MC count of the DCLNs were confirmed using a BX51 (Olympus, Tokyo, Japan) phase contrast microscope. The degranulation was expressed as a ratio by comparing the total number of MCs with the number of degranulated MCs.

### 4.10. Cell Culture

BV2 microglial cells were kindly donated by Professor Gi-Sang Bae (Wonkwang University, Iksan, Republic of Korea). The cells were cultured in the Roswell Park Memorial Institute-1640 medium (Gibco, Billings, MT, USA), supplemented with 10% fetal bovine serum (Gibco, USA) and 1% penicillin/streptomycin (100 U/mL, Gibco, USA), and incubated at 37 °C and 5% CO_2_. Cells were grown to 70–80% confluency and used in the experiments [63].

### 4.11. Cell Viability Assay

BV2 microglial cells were seeded in a 24-well plate at a density of 2 × 10^5^ cells/well [69]. Then, various concentrations of *Cannabis sativa* L. extracts (200, 400, 800, 1600, and 3200 µg/mL) were added, and cells were incubated for 24 h. Thereafter, 3-(4,5-dimethylthiazol-2-yl)-2,5-diphenyl-2H-tetrazolium bromide (MTT, Sigma, St. Louis, MO, USA) solution was added for 30 min. After removing the supernatant, dimethyl sulfoxide (DMSO, Sigma, St. Louis, MO, USA) was added to dissolve formazan. The solution was loaded into a 96-well plate, and absorbance was measured at 540 nm.

### 4.12. NO Assay

The NO concentration was determined using the Griess reagent [69]. BV2 microglial cells were seeded in a 24-well plate at a density of 2 × 10^5^ cells/well, pretreated with the *Cannabis sativa* L. extracts (200, 400, 800, and 1600 µg/mL) for 1 h, and co-treated with LPS (1 µg/mL) for 24 h. Equal volumes of the supernatant and Griess reagent were mixed at room temperature. Finally, the absorbance of the reaction mixture was measured at 540 nm using a spectrophotometer.

### 4.13. Real-Time Reverse Transcription Polymerase Chain Reaction (RT-PCR)

In order to extract total RNA, BV2 microglial cells were seeded at a density of 1 × $10^{6}$ cells/well in a 6-well plate [63,69]. *Cannabis sativa* L. extracts (200, 400, 800, and 1600 µg/mL) were pretreated for 1 h and co-treated with LPS (1 µg/mL) for 6 h. Then, the supernatant was removed, and the cells were lysed using the Easy-Blue RNA extraction kit (iNTRON Biotechnology, Sungnam, Republic of Korea). RNA purity was confirmed using a Gene Quant Pro RNA Calculator (Biochrom, Inc., Cambridge, UK). RNA was reverse-transcribed to cDNA using the ReverTra Ace qPCR RT Kit (Toyobo, Osaka, Japan). SYBR quantitative RT-PCR was performed using the ABI StepOne Plus detection system (Applied Biosystems, Thermo Fisher Scientific, Inc., Waltham, MA, USA), according to the manufacturer’s instructions. The PCR cycling conditions were as follows: 95 °C for 3 min and 45 cycles of 95 °C for 10 s, 60 °C for 10 s, and 72 °C for 20 s. For each sample, a triplicate test and a control reaction without reverse transcriptase were analyzed to evaluate the expression of the gene of interest and control variations in the reactions (Table 1).

### 4.14. Statistical Analysis

Based on the average value, the results were expressed as the mean ± standard deviation. All experimental results were analyzed using a one-way analysis of variance (ANOVA) with SPSS for Windows (version 26.0; IBM Corp., Armonk, NY, USA). The level of significance was set at *p* < 0.05.

## 5. Conclusions

*Cannabis sativa* L. leaf and inflorescence extracts contain a variety of bioactive phytochemicals, such as cannabinoids and terpenes. Through in vivo experiments, we showed that *Cannabis sativa* L. leaf extract administration improves anhedonia and depressive behavior, restores oxygen levels lowered by LPS administration, and inhibits MC degranulation in DCLNs. Additionally, through in vitro experiments, we found that *Cannabis sativa* L. leaf and inflorescence extracts exhibit anti-inflammatory activity by inhibiting pro-inflammatory cytokines and inflammatory mediators. We suggest that the efficacy of the *Cannabis sativa* L. leaf and inflorescence extracts identified in this study is due to the entourage effect of various bioactive phytochemicals. The results of our study suggest that *Cannabis sativa* L. leaf and inflorescence extracts have the potential to be an effective treatment for a variety of diseases associated with inflammatory responses. However, we are required to conduct future experiments such as Western blotting and immunofluorescence to investigate the mechanisms of *Cannabis sativa* L. leaf and inflorescence extracts’ antidepressant-like effects and anti-inflammatory activity.

## Figures and Tables

**Figure 1 plants-13-01619-f001:**
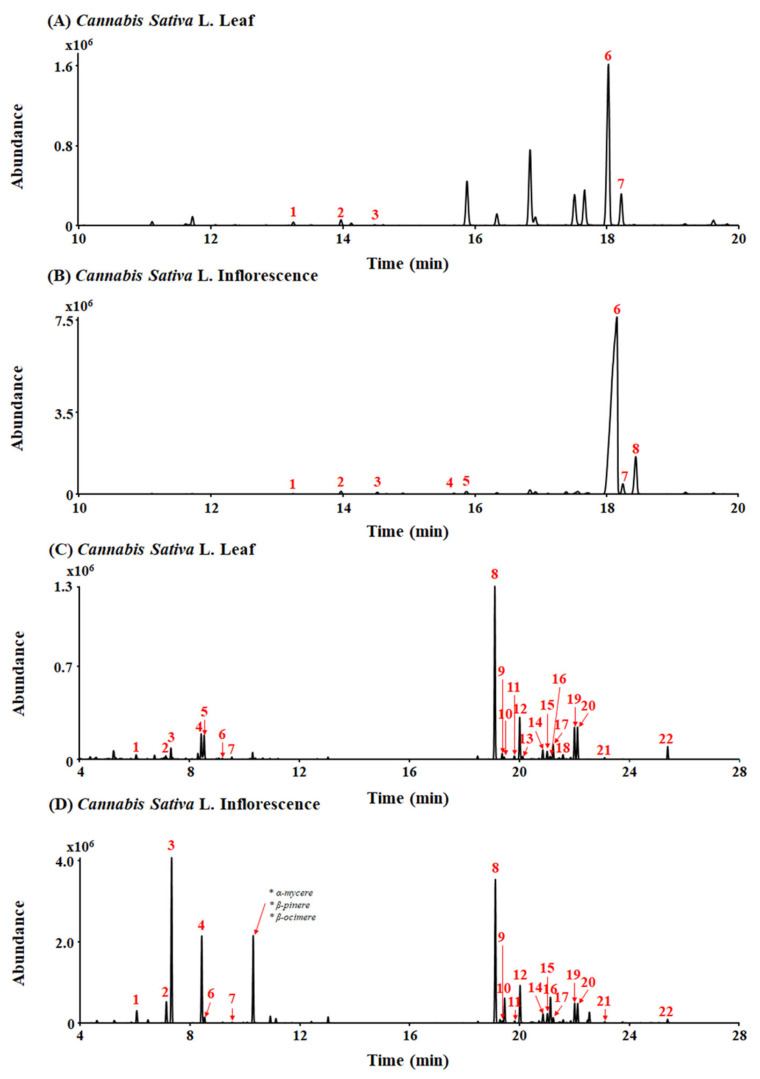
Total ion chromatograms (TICs) of leaf and inflorescence extracts of *Cannabis sativa* L. (**A**,**B**) Peak identities by GC-MS are as follows: 1. CBC; 2. Δ9-THC; 3. CBG; 4. CBDA; 5. THCVA; 6. THCA; 7. CBCA; and 8. CBGA. (**C**,**D**) Peak identities by headspace GC/MS are as follows: 1. α-pinene; 2. β-pinene; 3. myrcene; 4. limonene; 5. eucalyptol; 6. γ-terpinene; 7. Z-sabinene hydrate; 8. β-caryophyllene; 9. α-bergamotene; 10. α-guaiene; 11. E-β-farnesene; 12. α-humulene; 13. alloaromadrene; 14. β-selinene; 15. α-selinene; 16. Z,E-α-farnesene; 17. β-bisabolene; 18. β-sesquiphellandrene; 19. E-α-bisabolene; 20. selina-3,7(11)-diene; 21. caryophyllene oxide; and 22. α-bisabolol.

**Figure 2 plants-13-01619-f002:**
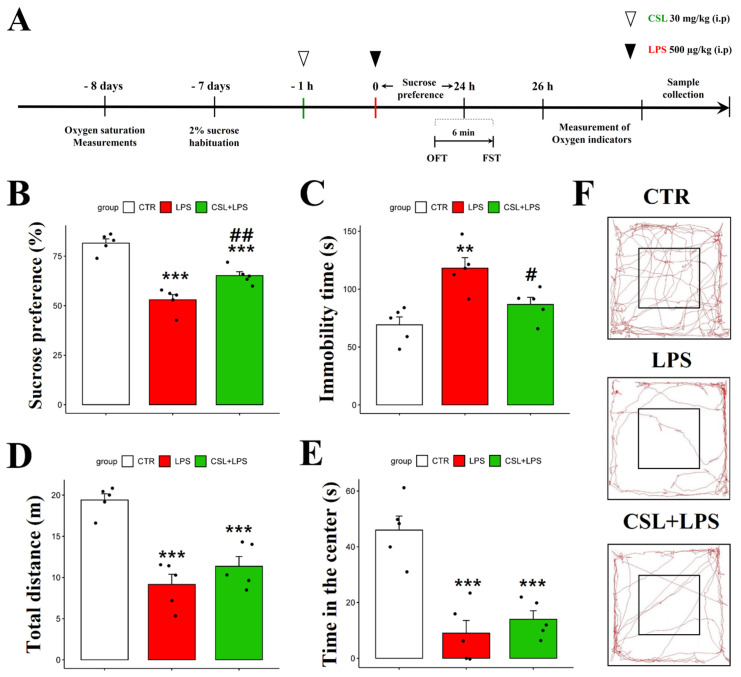
Behavioral effect of *Cannabis Sativa* L. (CSL) leaf extract administration in the LPS model. The experimental procedure (**A**), sucrose preference test (**B**), immobility time in the forced swimming test (**C**), total distance traveled in the OFT (**D**), time in the center in the OFT (**E**), and representative tracks of mice in the OFT (**F**). The groups were the control group (CTR), the LPS group, and the CSL+LPS group. Results are expressed as mean ± S.D., with individual data points shown as black dots. *, compared with the control group; #, compared with the LPS group. #, *p* < 0.05; **/##, *p* < 0.01; ***, *p* < 0.001 (*n* = 5 per group).

**Figure 3 plants-13-01619-f003:**
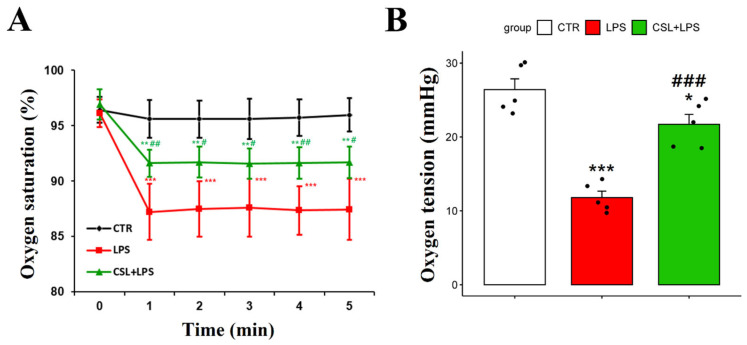
Changes in arterial oxygen saturation and oxygen tension following administration of *Cannabis sativa* L. (CSL) leaf extract. (**A**) The arterial oxygen saturation value of each group was measured every minute for a total of 5 min. (**B**) Oxygen tension of DCLNs measured with oxygen needle sensor. The groups were the control group (CTR), the LPS group, and the CSL+LPS group. Results are expressed as mean ± S.D., with individual data points shown as black dots. *, compared with the control group; #, compared with the LPS group. */#, *p* < 0.05; **/##, *p* < 0.01; ***/###, *p* < 0.001 (*n* = 5 per group).

**Figure 4 plants-13-01619-f004:**
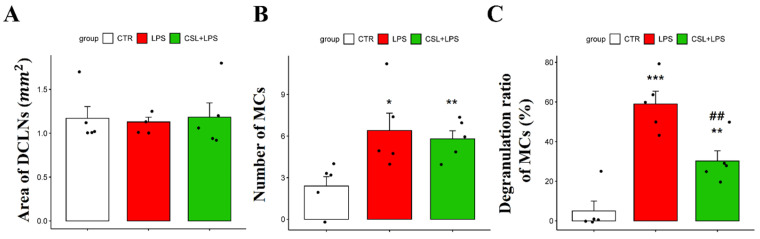
Effects of *Cannabis sativa* L. (CSL) leaf extract on MCs in DCLNs. (**A**) Area of DCLNs. The area was expressed as long axis × short axis × π. (**B**) Changes in the number of MCs and (**C**) the degranulation ratio of MCs. The groups were the control group (CTR), the LPS group, and the CSL+LPS group. Results are expressed as mean ± S.D., with individual data points shown as black dots. *, compared with the control group; #, compared with the LPS group. *, *p* < 0.05; **/##, *p* < 0.01; ***, *p* < 0.001 (*n* = 5 per group).

**Figure 5 plants-13-01619-f005:**
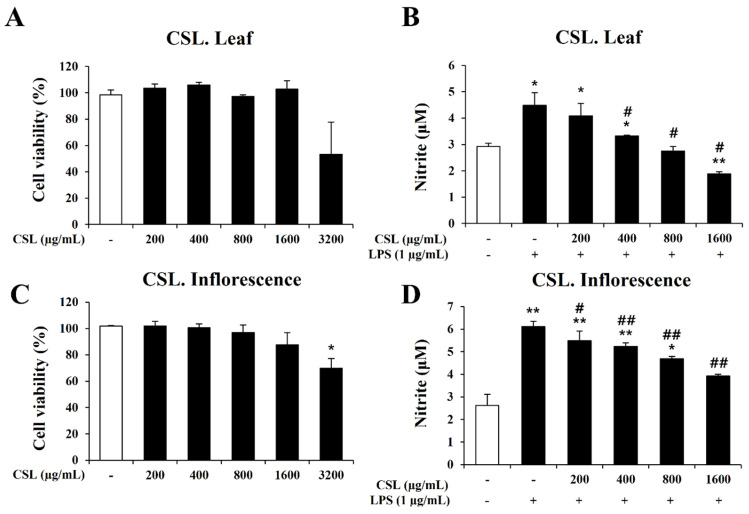
Effects of *Cannabis sativa* L. (CSL) leaf and inflorescence extracts on cell viability and NO production in BV2 cells. (**A**,**C**): Up to a dose of 1600 µg/mL, CSL leaf and inflorescence extracts did not affect cell viability. (**B**,**D**): The increase in nitrite by LPS was significantly decreased by CSL leaf and inflorescence extracts in a concentration-dependent manner. Results are presented as the mean ± standard deviation (S.D.). Results are representative of three independent experiments. *, compared with the saline treatment alone; #, compared with the LPS treatment alone. */#, *p* < 0.05; **/##, *p* < 0.01.

**Figure 6 plants-13-01619-f006:**
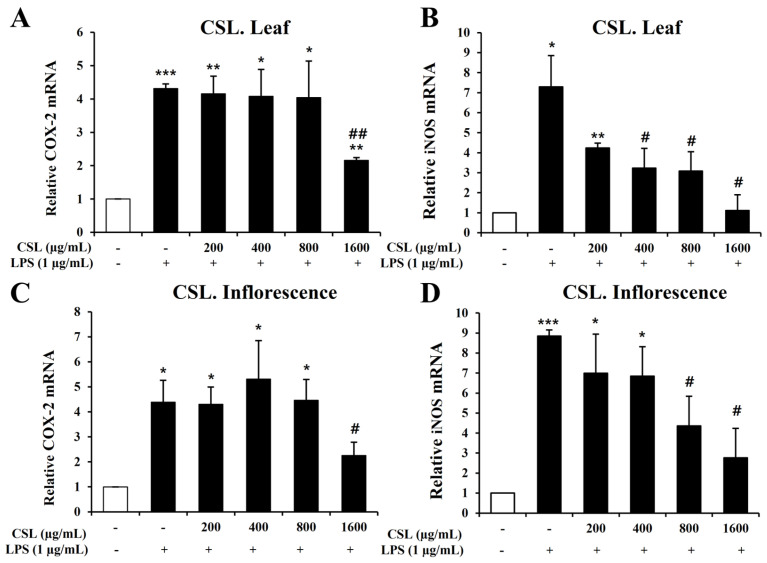
Effects of *Cannabis sativa* L. (CSL) leaf and inflorescence extracts on mRNA expression levels of iNOS and COX-2 in LPS-stimulated BV2 microglial cells. The CSL leaf extract effectively suppressed the mRNA levels of COX-2 and iNOS, particularly at 1600 µg/mL (**A**,**B**). The CSL inflorescence extract effectively suppressed COX-2 and iNOS mRNA levels in a concentration-dependent manner (**C**,**D**). Results are presented as the mean ± standard deviation (S.D.). Results are representative of three independent experiments. *, compared with the saline treatment alone; #, compared with the LPS treatment alone. */#, *p* < 0.05; **/##, *p* < 0.01; ***, *p* < 0.001.

**Figure 7 plants-13-01619-f007:**
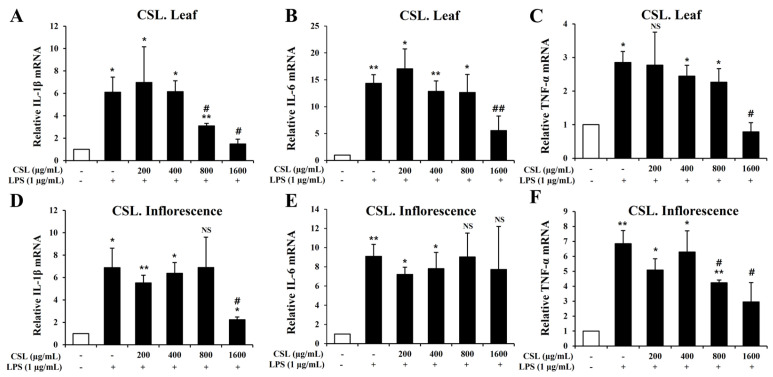
Effects of *Cannabis sativa* L. (CSL) leaf and inflorescence extracts on mRNA expression levels of IL-1β, IL-6, and TNF-α in LPS-stimulated BV2 microglial cells. The mRNA levels of IL-1β, IL-6, and TNF-α were suppressed in a concentration-dependent manner by the CSL leaf extract. This suppression was particularly effective at a concentration of 1600 µg/mL (**A**–**C**). Similarly, the CSL inflorescence extract also suppressed the mRNA levels of IL-1β and TNF-α, especially at a concentration of 1600 µg/mL (**D**–**F**). Results are representative of three independent experiments. *, compared with the saline treatment alone; #, compared with the LPS treatment alone. */#, *p* < 0.05; **/##, *p* < 0.01, NS; not significant.

**Table 1 plants-13-01619-t001:** Sequences of primers used for PCR.

Gene	Primer
IL-1β (F)	5′-CCT CGT GCT GTC GGA CCC AT-3′
IL-1β (R)	5′-CAG GCT TGT GCT CTG CTT GTG A-3′
IL-6 (F)	5′-CCG GAG AGG AGA CTT CAC AG-3′
IL-6 (R)	5′-CAG AAT TGC CAT TGC ACA AC-3′
TNF-α (F)	5′-AAC TAG TGG TGC CAG CCG AT-3′
TNF-α (R)	5′-CTT CAC AGA GCA ATG ACT CC-3′
iNOS (F)	5′-GTT GAA GAC TGA GAC TCT GG-3′
iNOS (R)	5′-GAC TAG GCT ACT CCG TGG A-3′
COX-2 (F)	5′-GGT GGC TGT TTT GGT AGG CTG-3′
COX-2 (R)	5′-GGG TTG CTG GGG GAA GAA ATG-3′
GAPDH (F)	5′-TGT GTC CGT CGT GGA TCT GA-3′
GAPDH (R)	5′-TTG CTG TTG AAG TCG CAG GAG-3′

## Data Availability

Data are contained within the article.

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
