# Peer review of "In Vitro and In Vivo Anti-Inflammatory and Antidepressant-like Effects of Cannabis sativa L. Extracts"

_plants, 2024, doi:10.3390/plants13121619_

Round 1

Reviewer 1 Report

Comments and Suggestions for Authors

Dear authors, the paper is good well written, but there some aspects that need to be corrected.

In abstract results must be written in the past

The introduction is good, please improve it by adding the traditional use of cannabis, also wehre the plant is growing

In material and methods how can you justify the dose 30mg/kg

The toxicity assesement was not done in the current study ? how can we know that it didn’t have alterations

In M&M groups were not put please add them and detail them please

Results

In fug 1 A pease correct the unit of LPS

As in M&M groups werenot clearly defined I think if a group that received only CSL would be very interesting

In fig 2 legend please write the p value in lower case ( and also other legends)

If it is possible to restrcure the paper by starting by in vitro then in vivo assesments

Discussion

Please use the past when describing results

In discussion no need to write figures ex in line 253,259,261… ‘’ in the center in the OFT (Figure 2D-F) ‘’

Also write the study limitations 

Reviewer 2 Report

Comments and Suggestions for Authors

This study focused on the in vitro and in vivo anti-inflammatory and antidepressant-like effects of Cannabis sativa L. extracts. The findings indicate that these extracts have the potential to be used as effective treatments for a variety of diseases associated with acute inflammatory responses. The topic of this study is interesting and fully fills in the scope of Foods. This study was designed logically and all of the results were good. Several suggestions for further improving the quality of the manuscript are shown below.

1.     The abstract should be improved. Important results (data) should be added, and the methods should be reduced in the abstract.

2.     For the extraction of Cannabis sativa L. leaf and inflorescence, are there any references ? The extraction method should be optimized.

3.     For the animal study, why 30 mg/kg was used in this study?

4.     Section 2.1, Total ion chromatograms of standards should be added, and the content of each identified compound should also be added. The author only provided the contents of β-caryophyllene in both Cannabis sativa L. leaf and inflorescence extract.

5.     The correlation analysis of individual compounds to the anti-inflammatory and antidepressant-like effects of Cannabis sativa L. extracts should be carried out.

Reviewer 3 Report

Comments and Suggestions for Authors

This paper is generally well written and presented and the experiments reasonably well planned.

Specific comments:

1.  L 85 “…was shown…” to “…”is shown…”

2.  The figure 1A peak just before 16 min appears to match peak 5 in 1B.

3. In figure 1 A & B peak 6 do not appear to have matching retention times even taking into account the fronting profile of peak 6 in 1B. The authors could confirm peak identities by comparing superimposing the MS for the peaks and comparing the m/z’s obtained and their relative intensities. 

4. In the label of figure 1C & D, they should be labeled as from head-space.

5. In figures 2 to 4, an explanation should be provided as to what the black dots represent and their significance in the vertical and horizontal positions.

6. In L 96-7, do the quantitation results refer to dry or wet weight and if dry weight, how was the sample dried and was the moisture content determined?

7. In L 140, what is the reason for the high standard deviations observed for oxygen saturation?

8. In section 4.8, how accurate is the pO2 sensor and was it calibrated just prior to use?

9. The unit “hour” is sometimes fully spelled out and sometimes abbreviated as “h”.

Reviewer 4 Report

Comments and Suggestions for Authors

The manuscript was well written. The authors analyzed the bioactive components in CannabisSatuva L. extracts. 

However, the effects of CSL in the LPS model did not fully support its antidepressant-like claim—especially the behavioral results. 

 In the behavioral study, how was the dose of 30 mg/kg CSL determined? Do you consider dosing CSL in various concentrations to determine the ED50? 

Are you considering extracting from various sources to determine any potential difference in the ratio of major active components? And its impact on anti-inflammatory and antidepressant-like effects? 

Comments on the Quality of English Language

English is fine.
